# Nutritional Care for Institutionalized Persons with Dementia: An Integrative Review

**DOI:** 10.3390/ijerph20186763

**Published:** 2023-09-15

**Authors:** María Elisa Moreno-Fergusson, Gabriela Rabe Caez-Ramírez, Luz Indira Sotelo-Díaz, Beatriz Sánchez-Herrera

**Affiliations:** 1Nursing and Rehabilitation School, Campus Puente del Común, Universidad de La Sabana, Chía 250001, Colombia; mariae.moreno@unisabana.edu.co; 2Engineering School, Campus Puente del Común, Universidad de La Sabana, Chía 250001, Colombia; gabriela.caez@unisabana.edu.co; 3EICEA Department of Gastronomy, Campus Puente del Común, Universidad de La Sabana, Chía 250001, Colombia; indira.sotelo@unisabana.edu.co

**Keywords:** dementia, Alzheimer’s, nutrition, meal environment, eating performance, feeding methods, nursing care

## Abstract

Background: Older people are at risk of malnutrition, especially when they suffer from cognitive impairment. Guidelines that orient nursing care in this regard need to be updated. The aim of this review is to address the best available evidence on interventions that can benefit nutritional nursing care for institutionalized older adults with dementia. Methods: Integrative review using the Dimensions and Eureka search engines, and the PubMed, Embase, Scielo, CINAHL, and ScienceDirect databases. We searched from the year 2015 through to 2021. We employed the MMAT guidelines for mixed, qualitative, and quantitative studies, and the PRISMA, CASP, and JBI guidelines to value the reviews. Results: A total of 55 studies met the inclusion criteria. The best available evidence to support nutritional nursing care for institutionalized older adults with dementia highlights several aspects related to the assessment and caring interventions that are focused on people with dementia, their caregivers, and their context. Conclusions: Both the assessment and nutritional care interventions for older people with dementia should consider the patient–caregiver dyad as the subject of care and understand the context as a fundamental part of it. The analysis of the context should look further than the immediate environment.

## 1. Introduction

Dementia is one of the leading causes of disability and dependence among older people worldwide [1].

It is related to people’s nutritional status in two ways: on the one hand, the possible association between poor nutrition and cognitive decline, and on the other hand, malnutrition occurring in people whose cognitive function is already impaired. In the first case, research indicates that some diets may help protect people from cognitive decline. For example, the Mediterranean diet, the DASH (Dietary Approaches to Stop Hypertension) diet, and balanced diets rich in protein are associated with less cognitive decline [2,3]. In the second case, impaired functionality in activities of daily living, characteristic of dementia, is associated with greater impairment in nutritional status [4,5]. Malnutrition in people with dementia has also been linked to an increased burden of care for caregivers [1].

Dementia affects a person’s ability to obtain a healthy diet, i.e., a diet that is sufficient, complete, and balanced to meet the body’s needs [1]. In older adults, dementia has different origins, although Alzheimer’s disease accounts for 70% of cases [1]. However, in Alzheimer’s dementia, frontotemporal dementia, dementia associated with Parkinson’s disease, Lewy body disease, progressive supranuclear palsy, Huntington’s dementia, or vascular origin dementia, there are relevant nutritional problems [6,7,8,9].

People with cognitive impairments have difficulty accessing food [10,11]. Moreover, when these individuals are institutionalized outside of their family environment, which may be long-term care homes or hospitals, they are more vulnerable to nutritional disorders [12,13].

In 2014, Guerchet et al. [14] presented an exhaustive analysis of the available evidence regarding nutrition in the cognitively impaired older adult, which addressed their context. However, reviewers at that time did not use an integrative approach that supported nutritional care for institutionalized older adults with dementia; therefore, it is necessary to make an updated review. Guerchet et al. established a relationship between the nutritional condition of older adults with dementia and their quality of life; they stated that although there is significant malnutrition in people with dementia, their nutritional status can be improved with dietary changes, interventions by the caregiver, and interventions in the care environment They also called for a more comprehensive approach to the nutritional health of these individuals, especially those who are institutionalized [14].

A better understanding of the interaction between nutritional health and cognitive status in institutionalized older adults could drive an improved quality of life for institutionalized older adults with early interventions, accurate treatments, and appropriate nursing care. This integrative review of the literature addresses interventions based on the best evidence available that can benefit the nutritional nursing care for institutionalized older adults with dementia.

## 2. Materials and Methods

An integrative review design was selected to address the interventions that can benefit the nutritional nursing care for institutionalized older adults with dementia. This type of review makes it possible to combine diverse study designs and data sources under a single framework while maintaining a rigorous and systematic approach. However, they demand a rigorous process with a defined search strategy, an evaluation of article quality, and a synthesis and conceptual analysis of the resulting information [15].

We conducted the search in October 2021, and used the Dimensions and Eureka search engines, and the PubMed, Embase, Scielo, CINAHL, and ScienceDirect databases. Search terms included “Dementia or Alzheimer’s disease”, “Nutrition or Food or Feeding Environment or Food Preparation or Feeding Methods or Culinary Medicine”, “Nursing Care or “Interventions”, and “Institutionalization”. Inclusion criteria consisted of studies and scientific documents in peer-reviewed publications published between 2015 and 2021. We included all languages and geographic areas. We excluded the gray literature documents, editorial material, expert opinions, and non-peer-reviewed or partially unavailable publications.

The researchers organized themselves into two independent groups to review and analyze the included articles critically. We used parameters according to the study methods of each document. Subsequently, we made a comparison between the quality concepts of the two researcher groups. When the concept diverged, the two researcher groups reviewed the process together to reach a consensus in the evaluation. In two cases, we contacted the authors to clarify specific research aspects of their papers. We employed the international recommended assessment tools for the quality evaluation of each type of study, to guarantee the rigor in the selection process that the researchers considered definitive to obtain the desired information: MMAT—Mixed Methods Appraisal Tool–guidelines [16] for mixed, qualitative, and quantitative studies, and the PRISMA [17], CASP [18], and JBI guidelines [19]. We also followed the Lobiondo and Haber parameters for grading the evidence level of each article [20].

We used a matrix to extract, summarize, and examine the data, including the Appendix A analysis, for the analysis of each study. This matrix included complete references, location of the study, objective, methodology, sample, evidence level, results, conclusions, and main contributions related to the search questions.

## 3. Results

Only 55 of the 470 studies met the selection criteria in this review (see Figure 1).

The 19 studies represented in this review were from 7 European countries; 17 were from North America; 10 from Asia; 2 from Oceania; and 7 were multicentric. We did not find any Latin American studies.

We found 8 level 1 studies, the highest possible level of evidence; 5 level 2 studies; 2 level 3 studies; 10 level 4 studies; 10 level 5 studies; 20 level 6 studies; and no level 7 studies, which coincides with the inclusion criteria.

We found 19 literature review studies, 7 qualitative studies, 6 secondary analyses, 3 methodological studies, 8 descriptive studies, 2 cohort studies, 3 correlational studies, 2 pilot studies, 1 clinical trial with a single arm, 2 randomized clinical trials with two arms, and 2 mixed methods studies (see Table 1).

The data synthesis allowed us to identify two inter-related categories that can benefit the nutrition of institutionalized people with dementia: caring assessment and caring intervention. Caring assessment must be a systematic process with subjective and objective parameters (overall clinical status and the context conditions). The other category is caring interventions that comprises nutrition care interventions focused on institutionalized persons with dementia, those focused on supporting their caregivers, and those focused on the institutional context where people receive their meals (see Figure 2).

### 3.1. Caring Assessment for Feeding and Nutrition of the Institutionalized Individuals with Dementia

According to the evidence, a detailed nutritional assessment is essential to guide the nutritional care of older adults with dementia who are institutionalized. This assessment should be person-centered, have subjective and objective parameters, consider the clinical condition, the institutional environment, and be properly supported [34,47].

First, it is necessary to assess the person’s nutritional history and any changes associated with advancing age or overall clinical status. Nutritional history is important because those who have had poor nutrition in the past are at an increased risk of nutritional imbalances [65]. Frequent changes that affect nutrition, such as anorexia or hyperphagia, decreased physical activity, and sleep disturbances are common in the elderly [8]. However, common changes such as sensory, gastrointestinal, metabolic, and functional decline; the presence of chronic diseases and polypharmacy; and dysphagia, electrolyte imbalances, and oral health should also be assessed [9,33,36,42,57]. Periodontal problems can affect the way of feeding and increase the risk of accidents in the feeding process. An oral health assessment should include a review of teeth, cavities, and prosthetic adjustments [33].

Second, it is crucial to assess the eating experience of institutionalized individuals, including their socialization skills. The findings of Anantapong et al. [22], Liu et al. [40], Martin et al. [8], McGrattan et al. [44], Palese et al. [47], and Wu et al. [66] indicate the importance of assessing ethnicity and culture, socioeconomic status, and context, with particular emphasis on companionship or isolation and institutional settings. Food preparation and storage [10,21], the level of social support or eventual motivation to participate in preparation [46], and economic constraints [65] add to the cultural and contextual factors of this experience. An evaluation of the social component should recognize the interpersonal and environmental relationships associated with eating as an essential aspect of the lived experience for these individuals, in addition to adequate food intake [13,36]. From a psychosocial perspective, emotional stress, loneliness, and mistreatment can affect eating behavior and nutrition [65].

Third, it is essential to assess the conditions of the place where the person eats. These include space, furniture, seating arrangements, noise, odors, lighting, routines, and the availability and presentation of food and utensils [4,5,13,25,26,28,34,36,38,39,46,50,57].

Fourth, it is also important to explore behaviors that influence nutrition, levels of food dependence, and the risks associated with food consumption. Eating behavior often helps to identify clinical features of dementia [61]. The authors identified four behavioral factors in patients with dementia related to alterations in weight and body mass index, food consumption, and low serum albumin levels. The first includes hypoactivity, drowsiness, and dietary agnosia. The second, hyperactivity, is related to agitation, wandering, rapid feeding, and delirium or delusion. Third are obsessive behaviors that include the refusal of food or faddish eating. Last are aberrant behaviors such as feeding apraxia, food theft, and ingestion of non-food objects. Agnosia and apraxia are common in people in the initial stages of dementia and make it difficult to identify familiar objects and perform appropriate sequential movements for feeding. Early detection of common symptoms of apraxia is part of the comprehensive assessment, as this is frequently associated with difficulty swallowing and increases the risk of aspiration pneumonia, dehydration, hospitalization, and death [6,48,64]. The authors of [6,48,64] recommend using the Kubota water swallowing test as a safe and simple assessment tool, in which the person drinks a spoonful of water and, if tolerated, drinks a glass of warm water.

Fifth, the assessment must be a systematic process, include the person with dementia, consider their caregivers [58], and use reliable tools under interprofessional supervision [55]. These people require a detailed analysis of functionality in different activities of daily living. One of the most used instruments is the Barthel Index, which classifies specific self-care skills such as drinking from a glass or cup, eating, dressing, grooming, bathing, sphincter control, and mobility functions 10. It is also necessary to determine the level of cognitive impairment. The Folstein Mini-Mental State Examination has proven its usefulness in this regard when used in conjunction with complementary tests that detect behavioral changes affecting nutrition [10]. It is also important to assess satiety, hyperorality, food storage in the mouth, or other behaviors that hinder proper nutrition in these individuals [5,61]. The Edinburgh Eating Assessment Scale in Dementia is useful to identify these behaviors and find out what support people need, through 10 simple questions [9].

Finally, the assessment should include objective parameters to establish and track nutritional status [28]. Anthropometric measures such as body mass, triceps skinfold thickness, arm circumference, serum albumin, and hemoglobin can complement the social and behavioral assessment [9].

### 3.2. Caring Interventions

Caring interventions are directed to either persons with dementia, their caregivers, or the context.

Nutrition care interventions focused on institutionalized persons with dementia.

Nutrition-related care interventions for institutionalized older adults with dementia aim to improve well-being or quality of life by preserving the best possible nutritional condition [4,32,38,49]. All these interventions try to respond to the needs and preferences of older adults as the primary beneficiaries [23,37].

It is recommended to propitiate positive food experiences that foster residual autonomy to improve well-being [4,9,25,26,27,29,34,37,38,42,45,48,50,51,54,56,57,59,62,66]. In this sense, interventions should make the most of subjects’ remaining abilities by providing verbal support, which is the key to motivating people to eat; visual support, i.e., role modeling; and partial physical support, such as putting food in utensils so that people can put the utensils in their mouths. Keller et al. [13] indicate that modifying psychosocial aspects while eating leads to improved food consumption and quality of life.

Interventions that favor the individual eating and nutrition process are especially important. Guidance while eating, careful food selection, and behavioral modifications can help people to eat more independently [11,29]. However, fully dependent individuals require complete assistance [9,10,24,57].

Keller et al. [13] recommend that chewing should be stimulated, and a pleasurable sensation should be maintained while eating. Herke et al. [28] suggest administering small portions at a slow pace to compensate for sensory deficits and maintaining proper positioning to help patients chew and swallow. It is also essential to modify the texture of food for people with dysphagia [33]. The eating behavior of these people is affected positively by Montessori-based activities that have been shown to favor nutrition, although they do not improve cognitive status [56].

Complementary measures may be useful in some cases. As noted by Tangvik et al. [62], Poscia et al. [50], and Saarela et al. [53], supplements such as vitamin D to prevent falls and fractures, or oral supplements and protein formulas to improve weight and decrease the risk of malnutrition, need to be included if necessary. In others, the assistance of a speech therapist in the personalized management of dysphagia may be of great support [11].

Nutritional care interventions focused on supporting caregivers.

Several interventions reported in the literature strive to avoid or alleviate the overload generated by the nutritional care of people with dementia on their family or professional caregivers [11,30,43,52,55,63].

Feeding-related processes are particularly stressful for caregivers, although they use different strategies to adapt to this condition [31,36]. However, the often-inappropriate behavior of people with dementia may cause them to become impatient, angry, desperate, anxious, or depressed [10,36]. The lack of information, knowledge, and support further hinders the situation [36]. Conversely, these caregivers have been shown to exhibit lower levels of depression when their caregiving skills improve [10].

It is critical for people with dementia to have caregivers with a caring and calm demeanor [28]. Shatenstein et al. [55] point out that the family caregiver must be involved to ensure that patients obtain their nutritional requirements. In this regard, it is important to have materials that clarify their nutritional requirements and give basic tips to adapt the person’s favorite dishes while ensuring a balanced diet. Thus, it requires managing uncertainty; reinforcing strategies, skills, and knowledge; and promoting an appropriate environment [11]. When caregivers participate in the design of interventions and are timely counseled about the disease trajectory, they can make better decisions, understand the goals of care, and avoid unnecessary interventions [27].

Programs such as case management, psychoeducational approach, counseling, support groups, respite care, psychotherapeutic approach, and multicomponent respite interventions have been effective in supporting caregivers [46]. Educational programs should address the psychosocial aspects of mealtimes and family relationships, coping, and stress management. Sharing experiences and reflecting on them is an effective way to increase understanding in complex situations like feeding people with dementia [29]. Some support strategies designed for caregivers, such as a voice app for managing nutrition problems, have been shown to facilitate decisions and reduce the perception of burden [35]. Mobile applications that aim to promote proper nutrition are more useful when they are easy to use and have detailed manuals [30].

For their part, formal caregivers face demanding complexities to ensure adequate nutrition for institutionalized persons with dementia. They require ongoing nutritional and dietary training to meet caregiving challenges [40]. Nutritional training for formal caregivers improves their knowledge, attitude, and behavior, and patient nutrition [43].

In addition to knowledge, these caregivers require time flexibility to help older adults eat, meet their individual demands, respond to their difficulties, and include their culture as a specific condition of their care plan [34]. In contrast, Featherstone et al. [12] point out that formal caregiving in highly controlled environments with strict schedules and routines lead to cycles of resistance that are associated with poor care and emotional and physical exhaustion. However, this field requires further exploration [36].

Nutritional care interventions focused on the institutional context where people receives their meals.

This group of interventions recommends that institutions specializing in the care of people with dementia, as well as their staff and food service providers, should focus on the individual’s environment to provide them with a healthy dietary intake [5,47,50]. The context is viewed as dynamic in these interventions [45].

It is essential to have organized processes to facilitate care with a focus on the environment. Snyder and Vitaliano (2020) recommend standardizing protocols, making time flexible, and being prepared. Models of care for the nutrition of these people, as proposed by Chen et al. [9] and Murphy et al. [45], have been useful to improve nutrition.

People with dementia tend to eat better when they can choose culturally appropriate foods, accompanied by a calm and familiar environment and adapted utensils [9,35]. A calm environment supports their concentration [46].

Rituals can facilitate eating by helping people identify when to eat [47]. Environmental adjustments including mealtimes, ringing a bell at mealtime, or opening the dining room can be helpful [37]. It is also necessary to plan the dining room layout and assign seating for a better social dining experience [38,47,66]. The use of appropriate utensils at the table, patient aprons, and proper seating posture, along with stimulating smells, sounds, and visual aids, have been shown to promote an appetite in these individuals [9,37]. Moreover, environmental modifications have shown improvements in anthropometric and functional indices [50].

Interventions that promote nutrition are dynamic and require adjustment, since dementia is a progressive disease with variable demands. Anantapong et al. [22] emphasize the relevance of a periodic assessment of nutritional status through clinical outcomes to guide these interventions. Palese et al. [47] suggest that the performance of food and the number of adverse events in individuals should also be monitored. In some cases, long-term measurements and the use of validated tools [41] may show better results. For example, three years after Soininen et al. [60] started their intervention, they found significant benefits in cognitive progression, function, and Alzheimer’s-related brain atrophy, with clinically relevant effect sizes not evident in short-term research.

## 4. Discussion

This review addresses interventions based on the best available evidence that could benefit the nutritional care of institutionalized older adults with dementia. The nutritional problems of these individuals are associated with their cognitive and metabolic status, neurochemical condition, and available support, which affect the individual and their caregiver [6].

The results of this review reflect the importance of a comprehensive view of the person with the dementia patient–caregiver dyad, immersed in a particular and unique context to develop care interventions focused on ensuring the feeding and nutrition processes.

Context is the physical, situational, political, historical, cultural, or any other environment in which an event takes place; everything that places people in a particular place and time, thus giving a specific meaning to the subject’s experience [67]. Therefore, it can be inferred that people perceive the context in a particular and unique way based on the association it makes with their previous experiences. In this way, the context constitutes an external and internal stimulus for the nourishment and nutrition of the elderly person with dementia and is, therefore, a determining factor in their care [28].

Environmental psychologists like Bronfenbrenner and Lawton emphasize the importance of the progressive adaptation of the person to the changing environment in which they live and develop [68,69]. It is essential to reconcile the interaction of older adults with the environment for their care, that is, to adapt the specific context in which they live, alongside their individual conditions, to create an environment that ensures their safety and general well-being considering their health state [69].

Based on the results of this review, we can identify some essential characteristics of interventions focused on food and nutrition: they are complex, interdisciplinary, dynamic, contextualized, and person-centered [45]. Their character is not linear; therefore, they require a shared analysis and a staged development [22,32] to meet the changing needs of the person with dementia and their caregiver [23].

The two categories that emerge from the analysis and synthesis of this integrated review must guide the nutritional care for institutionalized persons with dementia: caring assessment for feeding and nutrition of the institutionalized individuals living with dementia, and care interventions (see Figure 2).

The first identified category, caring assessment for feeding and nutrition of the institutionalized individuals living with dementia, includes the assessment of their health status and functionality of older adults with dementia: the risk or presence of dysphagia [64], eating behaviors, caregiver overload, and the physical and psychosocial environment of the dining room. The assessment of individuals with dementia provides critical information needed to establish their health condition [8], level of physical activity [42], and their functionality to perform activities of daily living, sleep, rest, and eating habits. These aspects are fundamental in the planning of interventions required to ensure food and nutrition [31], prevent adverse events, and avoid caregiver overload [35].

It is also important to assess feeding behaviors, the assisted feeding techniques they require, and to determine the contextual conditions that must be maintained or modified to ensure their well-being [9]. This is why assessments should be periodic: to adjust interventions to changing conditions, and include key aspects such as cultural traits, personal beliefs, and social norms [9].

The assessment of caregivers becomes very important if we consider that the main barriers and facilitators to eating are related to their caregiving competencies [40]. Among these are the lack of motivation, knowledge, preparation, and clinical training, and the work context which can generate competing work demands, time pressures, and frustration. Facilitators include competencies (knowledge, skills, and attitudes) and the motivation of staff to ensure quality care for people with dementia [40].

The second category, caring interventions, includes nutrition care interventions focused on institutionalized persons with dementia, on supporting caregivers, and on the institutional context where people receive their meals. A central aspect of this category is to guarantee an adequate diet, rich in protein and low in carbohydrates and sugars, to preserve cognitive capacity as much as possible [39]. Balanced diets that provide protein-based energy improve body weight and muscle strength [55], which leads to an improved mobility, functionality, and quality of life for people with dementia [54]. Diets can be supplemented with oral supplements and protein-based formulas [50,62] to reduce the risk of malnutrition, although there is a need for more robust studies to validate this statement.

Interventions focused on people with dementia are given in context and include the physical environment, psychosocial, normative, and cultural context [5]. In most of the studies found, the physical environment focuses on the dining room environment, and considers lighting, temperature, humidity, ventilation, visual contrast, aromas, sound control, routines, location of diners, utensils, preparation, and quality of food [9,29]. Considering that the environment is for the older adult part of their identity, it is very important that these interventions are customized to meet the needs of individuals [37].

The psychosocial context is given by the transactions that individuals make in their environment, such as the interaction with others (residents, family members, and health personnel) and their perception of support [46]. In this context, it is essential to simplify the instructions given to the person and to adapt the intervention to the patient’s tastes, preferences, and pace of mealtime, and to maintain a familiar atmosphere in the dining room [4,29]. In this process, it is mandatory to guarantee respect for the person’s dignity, autonomy, customs, personal beliefs, and social norms that have marked their lives.

Interventions aimed at caregivers are intended to strengthen their caregiving skills, ensure their well-being and that of the person with dementia, and reduce their perceived burden. According to Quinn et al., helping caregivers to find meaning and satisfaction in the performance of their role creates well-being and improves their quality of life [52]. They benefit when they have knowledge and receive training regarding feeding and nutrition, because it strengthens their empowerment, decision-making, and problem-solving skills while reducing uncertainty [11], and creates positive effects for the nutrition of patients [43,57]. Caregivers face an important challenge [12]. Nonetheless, feeding and nutrition can also be an opportunity to improve health, and strengthen bonds with patients through positive aspects of this experience [52]. However, caregivers need orientation to obtain these benefits [40].

The institutional context refers to the operating policies and norms of the institution, and their relationship with the design of menus, food preparation, and the cultural values of the people. In the normative context, the strict schedules of the institutions frequently gather resistance among residents and bring about changes in their behavior, which causes emotional and physical exhaustion for the staff [12]. It is important to consider the cultural values of people when designing menus for people with dementia [53].

In general terms, there are few studies that address the dyad as a subject of care in individuals’ assessments and interventions. Their description of the context is usually limited to the immediate physical, psychosocial, institutional, and cultural environment, without considering other areas described by Bronfenbrenner et al. [68], Lawton, and other authors with similar viewpoints [69]. Bronfenbrenner et al.’s ecological theory explains how individuals relate permanently and actively with their context in different dimensions, from permanent interaction to the media and government entities, which, although distant, can generate a more sustainable impact over time [68]. In Lawton’s person–environment fit theory, he argues that people’s behaviors are shaped by the environment in which they have developed throughout their lives [69].

Based on the two theories mentioned above, and in the results of the literature analysis, it can be suggested that future nursing care interventions should address the issue of food and nutrition of elderly people with dementia from a holistic perspective, which should recognize that the context is part of the identity of older adults with dementia and their caregivers. This context should be considered under theoretical references that articulate all its components.

This review has certain limitations: first, this study’s findings did not identify studies in the Latin American context, which limits the scope for implementation in this region. Moreover, there is some publication bias because we excluded studies published before 2015, beginning our review with Guerchet et al.’s review of available research conducted with Alzheimer’s Disease International [14].

## 5. Conclusions

This analysis of challenges and opportunities regarding the adequate nutritional status of institutionalized people with dementia points towards the need for an integral nutritional evaluation, nutritional interventions focused on the person with dementia, support for their caregivers, and an adequate feeding environment within the institution. It is necessary to research more evidence to back nursing practice regarding the feeding and nutrition of institutionalized people with dementia. This will decrease uncertainty and contribute to better decisions in daily life situations, allowing nurses to meet clinical demands, improve the user experience, and improve care management through adequate planning and more effective processes. Finally, it is important to point out that the interdisciplinary participation in this field will solidify as the challenges and opportunities related to the nutrition and feeding of institutionalized people with dementia become clearer.

## Figures and Tables

**Figure 1 ijerph-20-06763-f001:**
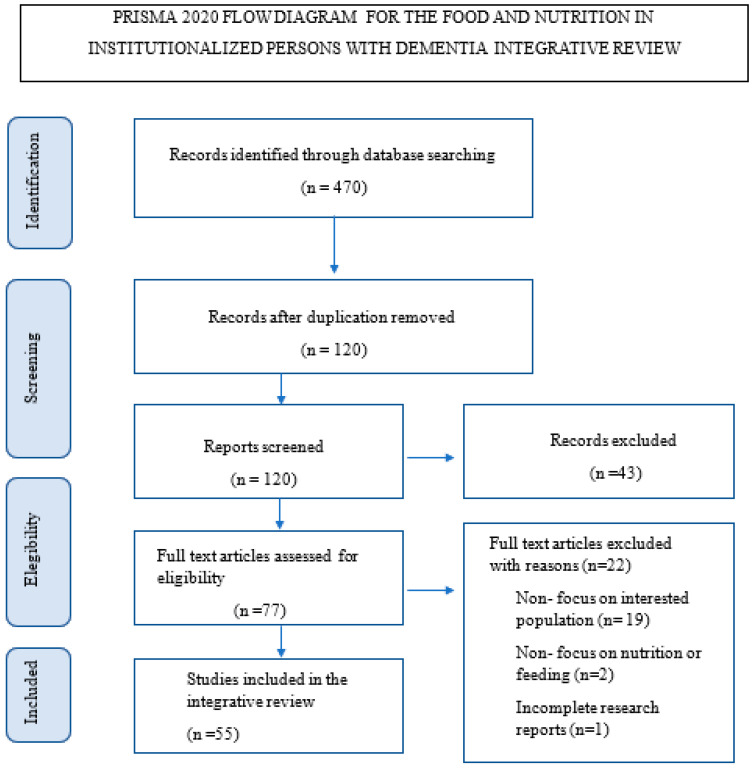
Flow chart of PRISMA 19.

**Figure 2 ijerph-20-06763-f002:**
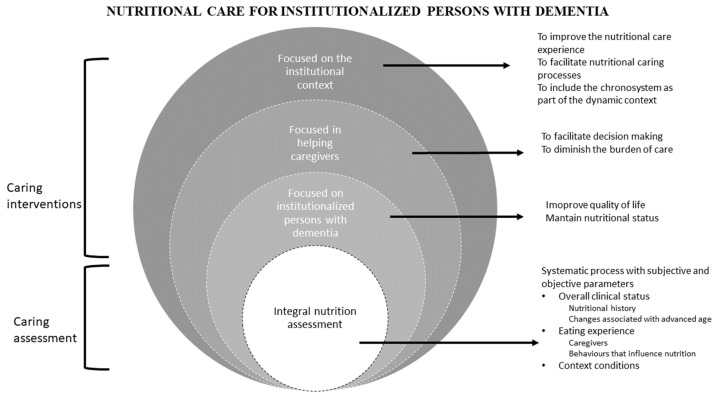
Categories of interventions.

**Table 1 ijerph-20-06763-t001:** Information about the included studies.

Author	Year	Title	Country	Method	Evidence	Main Results
Abdelhamid et al. [21]	2016	Effectiveness of interventions to directly support food and drink intake in people with dementia: systematic review and meta-analysis.	UK	Systematic review: *n* = 43.	1	They did not find definitive evidence of effectiveness, or lack of it on specific interventions. There is a need for more robust research.
Anantapong et al. [22]	2020	Mapping and understanding the decision-making process for providing nutrition and hydration to people living with dementia: a systematic review.	UK	Systematic review: *n* = 45.	1	Making decisions regarding nutrition and hydration for people living with dementia is not a linear process and demands a shared and stepwise manner support.
Arahata et al. [23]	2017	A comprehensive intervention following the clinical pathway of eating and swallowing disorder in the elderly with dementia: historically controlled study.	Japan	This is a single-arm, non-randomized trial: *n* = 90 patients.	3	The comprehensive geriatric assessment with multidisciplinary interventions enhanced dementia patients’ eating functional status.
Batchelor et al. [24]	2017	Experimental Comparison of Efficacy for Three Handfeeding Techniques in Dementia.	USA	Prospective pilot study with subjects’ experimental Latin square design and randomization to one of three handfeeding techniques: *n* = 30.	2	The 3 direct-, over-, and under-hand techniques were time neutral. Under- and direct-hand techniques helped to better diminish meal intake, feeding problematic behaviors.
Batchelor-M. et al. [25]	2019	Impact of Cognition and Handfeeding Assistance on Nutritional Intake for Nursing Home Residents.	USA	Secondary analysis research: *n* = 786 residents with and without dementia.	6	People with dementia that need assistance have a significantly lower food intake and therefore a higher risk of malnutrition. Snacks helped them increase their caloric intake.
Benigas, et. al. [26]	2016	Using Spaced Retrieval with External Aids to Improve Use of Compensatory Strategies During Eating for Persons with Dementia.	USA	Evaluative study on the effects of teaching from a baseline of compensatory swallowing behaviors (i.e., chin retraction): *n* = 5 patients.	4	Visual aid training strategies applied within an adapted context helped balancing 2–3 compensatory swallowing behaviors in each participant.
Bunn, et. al. [27]	2016	Effectiveness of interventions to indirectly support food and drink intake in people with dementia: Eating and Drinking Well IN dementia (EDWINA) systematic review. BMC geriatrics, 16(1), 1–21.	UK	Systematic review: *n* = 51 studies including 56 interventions.	1	There are several indirect interventions that may have a promising future to improve, maintain, or facilitate food and drink intake in people with cognitive impairment.
Chang et al. [10]	2017	Prevalence and factors associated with food intake difficulties among residents with dementia.	Taiwan	Correlational study: *n* = 213 residents with dementia.	4	The physical function and dining environment may be associated with food intake difficulties in people with dementia.
Chen et al. [9]	2016	Effects of a feeding intervention in patients with Alzheimer’s disease and dysphagia.	China	Prospective study: *n* = 30 Alzheimer’s patients with dysphagia.	4	Eating and feeding in patients with Alzheimer’s improved with a feeding intervention model that included patient, environment, food, and utensils preparation, as well as appropriate assistance, patient monitoring, psychological care, and nursing care after eating.
Cipriani et al. [6]	2016	Eating Behaviors and Dietary Changes in Patients with Dementia.	Italy	Literature review: *n* = 89.	5	Most dementia patients have several feeding and eating problems associated with their cognitive impairment, metabolic or neurochemical condition, or available care.
Faraday et al. [11]	2019	Equipping nurses and care staff to manage mealtime difficulties in people with dementia: A systematic scoping review of training needs and interventions.	UK	Systematic scoping review: *n* = 76.	5	Training needs are especially necessary to manage uncertainty. Training interventions on dysphagia management and communication are necessary.
Featherstone et al. [12]	2019	Routines of resistance: an ethnography of the care of people living with dementia in acute hospital wards and its consequences.	UK	Ethnography within 5 hospitals.	6	Cycles of resistance in response to organizationally imposed care schedules can result in poor care experiences for patients and lead to emotional and physical exhaustion for staff. More research is needed on how institutional goals can be better aligned to dementia patients’ needs.
Herke, M. [28]	2018	Environmental and behavioral modifications for improving food and fluid intake in people with dementia.	Germany	Systematic review: *n* = 9 studies (1502 patients with dementia).	1	Verbal and tactile communication from the caregiver can affect the nutrition of dementia patients by encouraging food intake. Objective measures of nutrition are indispensable to assess the effect of clinical trials. The environment is relevant as an external stimulus for the nutrition of the elderly person with dementia. The evidence on environmental interventions is weak.
Johansson et.al. [29]	2017	Staff views on how to improve mealtimes for elderly people with dementia living at home.	Sweden	Descriptive qualitative study with 4 focal groups. Participants included 22 workers that care for people with dementia.	6	The quality of meals for dementia patients can be improved with a person-centered approach. It is necessary to promote autonomy and maintain a family environment at mealtimes. It is also necessary to share staff experiences.
Jung et al. [30]	2020	Feasibility of a Mobile Meal Assistance Program for Direct Care Workers in Long-Term Care Facilities in South Korea.	S. Korea	Mixed study that follows 23 dyads of older adults with dementia and their attendants from a long-term care facility in South Korea.	4	The APP intervention may be useful in the long-term care of people with dementia, especially for formal caregivers who need training in meal attendant skills.
Kawaharada et al. [31]	2019	Impact of loss of independence in basic activities of daily living on caregiver burden in patients with Alzheimer’s disease: A retrospective cohort study.	Japan	Retrospective cohort study of people with Alzheimer’s disease: *n* = 117.	4	The turndown in functionality in activities of daily living related to bathing and feeding people with dementia are risk factors for the caregiver burden.
Keller et al. [13]	2018	Development and Inter-Rater Reliability of the Mealtime Scan for Long-Term Care.	Canada	Methodological study: *n* = 82 (in different dining rooms in 32 long-term care facilities).	6	The MTS scale showed facial validity and reliability to assess the physical and psychosocial environments in shared dining rooms during meals. It is necessary to test its constructs.
Keller, H. [32]	2016	Improving food intake in persons living with dementia.	Canada	Narrative review of the literature: *n* = 74 studies.	5	Malnutrition in people with dementia is associated with poor food intake. Modifying the psychosocial aspects during mealtimes may improve food intake and quality of life.
Kobayashi et al. [33]	2017	Prevalence of oral health-related conditions that could trigger accidents for patients with moderate-to-severe dementia.	Japan	Descriptive quantitative study: *n* = 92 dementia patients with oral problems.	5	Patients with moderate-to-severe dementia have a high prevalence of oral health conditions that increase the risk of accidents during mealtimes.
Leah, V. [34]	2016	Supporting people with dementia to eat.	UK	Systematic review: *n* = 22 studies.	1	Recommended interventions focused on education, environmental changes, routine management, and feeding support. There is a need to ensure that staff carry out individual assessments to identify people who are having difficulty eating and to guarantee that they have enough time to eat.
Li et al. [35]	2020	Overview of systematic reviews: Effectiveness of non-pharmacological interventions for eating difficulties in people with dementia.	China	Umbrella review: *n* = 18 systematic reviews.	1	A review of non-pharmacological interventions provides weak evidence that could help caregivers select the most effective strategies for coping with feeding difficulties and preventing adverse events.
Li et al. [36]	2020	Informal dementia caregivers’ experiences and perceptions about mealtime care: A qualitative evidence synthesis.	China	Metasynthesis: *n* = 10 articles.	5	Caregivers need to be knowledgeable and informed about services related to the feeding of people with dementia to care for them appropriately. The experiences and clinical skills of informal caregivers are relevant to improve the quality of mealtime care.
Liu et al. [4]	2016	Factors associated with eating performance for long-term care residents with moderate-to-severe cognitive impairment.	USA	Secondary analysis of data from 192 residents in 8 institutions.	6	Interventions aimed at reducing the impact of cognitive impairment and improving the physical capacity of residents are required to optimize feeding performance.
Liu et al. [37]	2017	The association of eating performance and environmental stimulation among older adults with dementia in nursing homes: A secondary analysis.	USA	Secondary analysis of videos taken from residents at lunchtime. There are 36 videos, 15 residents with dementia, and 19 nursing assistants.	6	An association was found between the dietary performance of elderly people with advanced dementia and associated comorbidities. Environmental stimulation should be personalized to respond to their unique conditions and preferences.
Liu et al. [38]	2019	Factors influencing the pace of food intake for nursing home residents with dementia: Resident characteristics, staff mealtime assistance and environmental stimulation.	USA	A secondary analysis: *n* = 36 videos.	6	Factors influencing mealtime pace include being male, having more interactions, having physical and visual assistance, and a better performance during feeding.
Liu et al. [39]	2020	Association between Intake of Energy and Macronutrients and Memory Impairment Severity in US Older Adults, National Health and Nutrition Examination Survey 2011–2014.	USA China Italy UK Canada	Survey research: *n* = 3623 participants older than 60 years.	4	The authors related the intake of a healthy diet with better cognitive capacity, pointing out that people who consume more carbohydrates and sugars have a higher risk of having memory impairment than those who consume more protein-based energy.
Liu et al. [40]	2020	Facilitators and barriers to optimizing eating performance among cognitively impaired older adults: A qualitative study of nursing assistants’ perspectives.	USA	Descriptive qualitative study in 2 hospitals with 6 focal groups that included 23 nurses.	6	The main barriers and facilitators of feeding performance are related to caregivers. According to them, these are related to physical, social, and cultural aspects of the environment, as well as to the norms of the institution. Barriers include lack of preparation and training, competing work demands, time pressure, and frustration. Facilitators involve preparation, motivational, technical, informational, and instrumental assistance. It is recommended that programs include staff training emphasizing individualized care.
Liu et al. [41]	2020	Ease of use, feasibility and inter-rater reliability of the refined Cued Utilization and Engagement in Dementia (CUED) mealtime video-coding scheme.	USA	Methodological study with a secondary analysis of 110 observations of video recordings: *n* = 25 residents and 29 assistants.	6	The CUED scheme for analyzing videos of food intake in interaction with the person and the dementia patient–caregiver dyad was preliminarily valid, reliable, and useful.
Liu et al. [42]	2021	Nutrition and exercise interventions could ameliorate age-related cognitive decline: a meta-analysis of randomized controlled trials.	USA	Systematic review and meta-analysis: *n* = 6 clinical randomized control trials with 1039 participants.	1	There is a beneficial effect of proper nutrition combined with exercise on the overall cognitive function of individuals.
Marples et al. [43]	2017	The effect of nutrition training for health care staff on learner and patient outcomes in adults: a systematic review and meta-analysis.	UK	Systematic review and meta-analysis of clinical trials of staff training: *n* = 24 studies.	1	Despite the weak evidence, it can be stated that nutrition training can benefit the knowledge, attitude, and behavior of caregivers of the vulnerable elderly with effects on the nutritional intake of patients with dementia.
Martin et al. [8]	2018	Body composition, dietary, and gustatory function assessment in people with Alzheimer’s disease.	Spain	Descriptive comparative, cross-sectional cohort study: *n* = 75 people with Alzheimer’s disease compared with 267 healthy older adults.	4	People with Alzheimer’s disease had significant differences. They had lower body mass indexes, more hours of sleep, greater alteration in the perception of sweet and salty tastes, and less adherence to exercise and to the Mediterranean diet. These differential factors increase the risk of malnutrition in those with Alzheimer’s disease.
McGrattan et al. [44]	2021	A mixed methods pilot randomized controlled trial to develop and evaluate the feasibility of a Mediterranean diet and lifestyle education intervention ‘THINK-MED’ among people with cognitive impairment.	UK	A mixed methods pilot randomized controlled trial: *n* = 20 participants.	2	The authors present the development and preliminary evaluation of the Think-MED intervention, which strives for adherence to a Mediterranean diet. The results are not conclusive.
Murphy et al. [45]	2017	Nutrition and dementia care: developing an evidence-based model for nutritional care in nursing homes.	UK	Qualitative study with 9 focus groups.	6	The authors present an empirical model with an overall theme of person-centered nutritional care and a collaborative approach. They add to this view the availability of food and drink, tools, resources, contact, interaction when eating and drinking, participation in activities, and consistency of care and information.
Nell et al. [5]	2016	Factors affecting optimal nutrition and hydration for people living in specialized dementia care units: A qualitative study of staff caregivers’ perception.	New Zealand	Descriptive qualitative study: *n* = 11 caregivers.	6	The factors that affect nutrition and hydration in people living with dementia are complex, inter-related, and include both personal and environmental aspects. Therefore, both aspects included must be considered within the care given to these people.
Palese et al. [46]	2018	Interventions maintaining eating Independence in nursing home residents: a multicenter qualitative study.	Italy	Qualitative descriptive multicenter study. Participants included 54 health professionals interviewed in 13 focus groups.	6	The experience of caring for people with dementia of the study participants adds new interventions at mealtime in addition to those already described. Documented interventions include individualized care, environmental, and social interaction.
Palese et al. [47]	2020	Enhancing independent eating among older adults with dementia: a scoping review of the state of the conceptual and research literature.	Italy UK	Scoping review with 17 documents included.	5	Recommended interventions to improve the diet of people with dementia must have an explicit conceptual framework and integrate individual, social, cultural, and environmental factors. When evaluating the effectiveness of these interventions, dietary performance, clinical outcomes, and adverse events should be considered.
Painter et al. [48]	2017	Texture-modified food and fluids in dementia and residential aged care facilities.	Australia	Literature review with 22 studies.	5	Fluoroscopy showed that texture-modified food reduced the risk of aspiration in people with dementia. These meals did not reduce clinical aspiration or pneumonia incidence and were associated with lower daily energy and fluid intake. Their adherence was variable.
Park et al. [49]	2018	National study of the nutritional status of Korean older adults with dementia who are living in long-term care settings.	Korea	Secondary analysis of data from the Nationwide Survey on Dementia Care in Korea: *n* = 3474 participants.	6	The nutritional status of older adults with dementia living in long-term care facilities in South Korea was found to be poor and associated with multiple factors. The malnutrition rate was 38% and the risk of this was 55%. Most of the malnourished older adult population with dementia was institutionalized and suffered from cognitive impairment, with functional dependence and associated comorbidities.
Poscia et al. [50]	2018	Effectiveness of nutritional interventions addressed to elderly persons: umbrella systematic review with meta-analysis.	Italy Poland	Umbrella systematic review with meta-analysis: 28 articles.	2	The nutrition of the elderly requires a preventive approach to improve their quality of life. Personalized nutritional interventions are suggested that consider functional conditions and cognitive status and are supported by solid evidence, such as vitamin D supplementation that helps prevent falls and fractures. In the same way, they suggest oral supplements and protein-based formulas that reduce the risk of malnutrition and improve weight.
Prizer et al. [51]	2018	Progressive Support for Activities of Daily Living for Persons Living with Dementia.	USA	Review of the gray literature: *n* = 59 references.	6	Person-centered care is recommended in which dementia is considered to require progressive support as functionality is lost. Care should consider the tastes and preferences of the person. Simple instructions are easier to follow. It is necessary to keep in mind the respect for dignity, try to preserve autonomy, respect customs, know the personal health situation, monitor the environment considering the necessary adaptations, and the characteristics of food and drinks and interaction at mealtime.
Quinn et al. [52]	2019	Influence of Positive Aspects of Dementia Caregiving on Caregivers’ Well-Being: A Systematic Review.	UK	Systematic review: *n* = 53 studies.	5	Identification of positive aspects of dementia care is associated with better caregiver well-being. Therefore, it is necessary to develop approaches and measures to promote them. Longitudinal studies are also required, as these experiences may change over time.
Saarela et al. [53]	2017	Changes in malnutrition and quality of nutritional care among aged residents in all nursing homes and assisted living facilities in Helsinki 2003–2011.	Finland	Retrospective observational study 2003 (*n* = 1987), in 2007 (*n* = 1377), in 2011 (*n* = 1576), and (4) in 2011 (*n* = 1585).	4	Institutionalized residents are at a higher risk of presenting nutritional problems. Institutions must improve their efforts to provide better nutritional care to the elderly.
Salminen et al. [54]	2019	Energy Intake and Severity of Dementia Are Both Associated with Health-Related Quality of Life among Older Long-Term Care Residents.	Finland	Descriptive correlational: *n* = 583 residents.	4	Energy intake has a correlation with HRQL in people with dementia. There is a positive correlation between these two variables, especially in the dimensions of mobility and daily activities in people with mild and moderate dementia, but not with those who have severe dementia.
Shatenstein et al. [55]	2017	Outcome of a Targeted Nutritional Intervention Among Older Adults with Early-Stage Alzheimer’s Disease: The Nutrition Intervention Study.	Canada	Intervention study: *n* = 67 dyads. Intervention group: *n* = 34 dyads and control group: *n* = 33.	3	The results show a higher intake of fat, energy, protein, and calcium in the intervention group, which improves body weight and muscle strength compared with the control group. Long-term follow-up is required to verify that these results are maintained over time.
Sheppard et al. [56]	2016	A Systematic Review of Montessori-Based Activities for Persons with Dementia.	Canada	Systematic review: *n* = 14 articles.	5	The results show the benefits of Montessori-based activities on eating behaviors but not on cognitive processes in people with dementia. Given the variety of the characteristics of the interventions, it is necessary to standardize the approach in future studies in order to analyze their benefits in the long term.
Simmons et al. [57]	2018	A Quality Improvement System to Manage Feeding Assistance Care in Assisted-Living.	USA	Observational study with *n* = 53 residents.	6	The implementation of the system to improve the quality of assistance during and between meals to institutionalized people with dementia showed positive results, evidenced in a reduction in weight loss.
Snyder et al. [58]	2020	Caregiver Psychological Distress: Longitudinal Relationships with Physical Activity and Diet.	USA	Correlational study: *n* = 122 caregivers of spouses with Alzheimer’s disease, 117 non-caregivers.	4	The results of this study highlight the effect of hours of care for people with Alzheimer’s and the psychological distress related to the health behaviors of caregivers.
Soininen et al. [59]	2017	24-month intervention with a specific multinutrient in people with prodromal Alzheimer’s disease (LipiDiDiet): a randomized, double-blind, controlled trial.	Finland Sweden Netherlands USA Germany	RCT *n* = 311 with individuals with prodromal Alzheimer’s disease.	2	The Fortasyn Connect intervention did not show a significant effect on the neurocognitive functions of the intervention group when compared to the control group. These researchers found group differences related to disease progression in cognitive and hippocampal function and cortical atrophy. Studies with larger sample sizes and the determination of a more sensitive endpoint in people with predementia are required.
Soininen, et al. [60]	2021	36-month LipiDiDiet multinutrient clinical trial in prodromal Alzheimer’s disease.		RCT: *n* = 311 with individuals with prodromal Alzheimer’s disease.	2	A period of 36 months’ follow-up showed the benefits of the Fortasyn Connect multinutrient intervention. The results show that the process of cognitive deterioration, functionality, and brain atrophy in people with Alzheimer’s disease slowed down. These results indicate that the benefits of the intervention increased with long-term use.
Takada et al. [61]	2017	Grouped factors of the ‘SSADE: signs and symptoms accompanying dementia while eating and nutritional status—An analysis of older people receiving nutritional care in long-term care facilities in Japan.	Japan	Methodological research: *n* = 259.	6	The results show that the four factors measured by the SSADE (hypoactivity, hyperactivity, obsession, aberrant behaviors) are related to nutritional status and have acceptable factorial validity. The first three factors are negatively correlated with BMI, and aberrant behaviors are positively correlated with dietary intake.
Tangvik et al. [62]	2021	Effects of oral nutrition supplements in persons with dementia: A systematic review.	Norway	Systematic review: *n* = 10 studies.	5	The intake of Oral Nutritional Supplements improves the nutritional status of people with dementia, which translates to an increase in body weight, muscle mass, and nutritional biomarkers. Studies do not show the impact of these supplements on the cognitive and functional level.
Tombini et al. [63]	2016	Nutritional Status of Patients with Alzheimer’s Disease and Their Caregivers.	Italy Canada	Descriptive: *n* = 90.	6	The results show a high prevalence of malnutrition in patients with Alzheimer’s disease associated with impaired functionality. In the caregivers group, 23% had malnutrition and 41.1% were at risk. This situation is associated with age, functionality, educational level, and cognitive status. It is imperative to establish nutrition education programs and policies to promote the nutrition of patients and caregivers.
Watanabe et al. [64]	2019	Association between dysphagia risk and unplanned hospitalization in older patients receiving home medical care.	Japan Taiwan Australia	Secondary analysis: *n* = 128.	6	The results of this study show that it is possible to predict the risk of unexpected hospitalization in older adults when they are at risk of dysphagia, hence, the importance of determining this risk in all home-care patients.
Whitelock et al. [65]	2018	On your own: older adults’ food choice and dietary habits.	UK	Qualitative focus groups: *n* = 30.	6	The authors identified changes associated with age, access to food, feeding on their own, and relationship with food, with 12 subthemes, which affect the acquisition and preparation of food and eating habits. Loneliness is an aspect that affects eating habits. It is important to develop policies and programs that favor the acquisition of food and opportunities for eating in the company of others.
Wu et al. [66]	2018	Mixed methods developmental evaluation of the CHOICE program: a relationship-centered mealtime intervention for long-term care.	Canada	Mixed methods research: *n* = 64 residents, 25 care staff/home management.	6	Participants in this study consider that the CHOICE (Connecting, Honoring dignity, Offering support, supporting Identity, Creating opportunities, and Enjoyment) multicomponent intervention is an appropriate strategy for managing mealtime challenges. The experiences with its application during meals were satisfactory.

## Data Availability

Data are available on request due to restrictions, e.g., privacy or ethical. The data presented in this study are available on request from the corresponding author. The data are not publicly available due to the group being interested in improving and making contact with different organizations that are interested in the same topics.

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
