# Peer review of "Nutritional Care for Institutionalized Persons with Dementia: An Integrative Review"

_ijerph, 2023, doi:10.3390/ijerph20186763_

Round 1

Reviewer 1 Report

The manuscript is very interesting; however it needs to be considered the following reviews first.

The introduction must better clarify the reason why, in the current state of knowledge, it is necessary to develop a previous integrated system.

A guideline or more systematic approach should be used in materials and methods to highlight and document how the supplementary review was accomplished.

For example, it is not clear why time periods were included by who performed the review and how double checks were implemented to ensure quality in the process.

It is not clear why the critical evaluations of the studies were carried out, using inappropriate tools if the results of these evaluations were not taken into consideration in the results, if not very limited.

Indeed, it is not clear why after the critical evaluation of the available evidence you have collapsed all the interventions into a single framework.

Precisely for the purpose of the integrative evaluation which does not necessarily require the critical evaluation of the studies, I would suggest that you consider the revision you have made as a basis for creating the reference framework without carrying out the passage of the critical evaluation with the different tools which appears very weak also from a methodological point of view and which is not used in the results. The discussion is very limited and needs to be expanded within the scope of the study. The study is interesting but the objectives need to be modified, the critical evaluation part of the studies removed and attention focused on interventions to improve and focus on the nutrition of these patients

Author Response

We appreciate the careful reading and suggestions to the article. Below we report how we dealt with each of your comments, which undoubtedly helped us to improve the article.

Reviewer 2 Report

The research question for this literature review is not clearly stated. On page 2, we are told that this review analyzes nutrition and feeding habits in institutionalized people with dementia. On page 9, the results are presented as 4 categories of interventions that can benefit the nutritional of institutionalized people with dementia. This is not the same as the research question. Was the research question to specifically address interventions that can benefit this group? This would be more aligned with the results.

Why was 2015 chosen as the year to start the search?

The discussion section is quite brief and would benefit from more detail. There are a number of paragraphs with only one sentence, and there are points that are repetitious from the background.

Overall this is an interesting topic to examine, and the points above could be easily addressed by the authors.

Author Response

(The authors gave the same response as above.)
